# Anticancer Activities of Novel Nicotinamide Phosphoribosyltransferase Inhibitors in Hematological Malignancies

**DOI:** 10.3390/molecules28041897

**Published:** 2023-02-16

**Authors:** Paulina Biniecka, Saki Matsumoto, Axel Belotti, Jessie Joussot, Jian Fei Bai, Somi Reddy Majjigapu, Paul Thoueille, Dany Spaggiari, Vincent Desfontaine, Francesco Piacente, Santina Bruzzone, Michele Cea, Laurent A. Decosterd, Pierre Vogel, Alessio Nencioni, Michel A. Duchosal, Aimable Nahimana

**Affiliations:** 1Central Laboratory of Hematology, Department of Medical Laboratory and Pathology, Lausanne University Hospital and University of Lausanne, 1011 Lausanne, Switzerland; 2Laboratory of Glycochemistry and Asymmetric Synthesis, Swiss Federal Institute of Technology (EPFL), 1015 Lausanne, Switzerland; 3Service and Laboratory of Clinical Pharmacology, Department of Laboratory Medicine and Pathology, Lausanne University Hospital and University of Lausanne, 1011 Lausanne, Switzerland; 4Department of Experimental Medicine, Section of Biochemistry, University of Genoa, 16132 Genoa, Italy; 5Department of Internal Medicine and Medical Specialties, University of Genoa, 16132 Genoa, Italy; 6Ospedale Policlinico San Martino IRCCS, Department of Internal Medicine, University of Genoa, 16132 Genoa, Italy; 7Service of Hematology, Department of Oncology, Lausanne University Hospital and University of Lausanne, 1011 Lausanne, Switzerland

**Keywords:** NAMPT inhibitor, NAD, anticancer, leukemia, lymphoma, multiple myeloma, ATP, apoptosis, oxidative stress, vitamin B3, PK studies

## Abstract

Targeting cancer cells that are highly dependent on the nicotinamide adenine dinucleotide (NAD+) metabolite is a promising therapeutic strategy. Nicotinamide phosphoribosyltransferase (NAMPT) is the rate-limiting enzyme catalyzing NAD^+^ production. Despite the high efficacy of several developed NAMPT inhibitors (i.e., **FK866** (APO866)) in preclinical studies, their clinical activity was proven to be limited. Here, we report the synthesis of new NAMPT Inhibitors, **JJ08**, **FEI191** and **FEI199**, which exhibit a broad anticancer activity in vitro. Results show that these compounds are potent NAMPT inhibitors that deplete NAD^+^ and NADP(H) after 24 h of drug treatment, followed by an increase in reactive oxygen species (ROS) accumulation. The latter event leads to ATP loss and mitochondrial depolarization with induction of apoptosis and necrosis. Supplementation with exogenous NAD^+^ precursors or catalase (ROS scavenger) abrogates the cell death induced by the new compounds. Finally, in vivo administration of the new NAMPT inhibitors in a mouse xenograft model of human Burkitt lymphoma delays tumor growth and significantly prolongs mouse survival. The most promising results are collected with **JJ08**, which completely eradicates tumor growth. Collectively, our findings demonstrate the efficient anticancer activity of the new NAMPT inhibitor **JJ08** and highlight a strong interest for further evaluation of this compound in hematological malignancies.

## 1. Introduction

Cancer cells have very high nutrient and energy demands in order to sustain their constant growth and rapid cell proliferation. Their metabolic reprogramming has recently emerged as an important cancer hallmark [1,2]. First described by Otto Warburg, a particular characteristic of cancerous cells resides in their preference towards aerobic glycolysis over oxidative phosphorylation [3]. However, this preference does not exclude the involvement of oxidative metabolism. Malignant cells rely on ATP and oncometabolite production from the mitochondrial tricarboxylic acid (TCA) cycle as much as they rely on glycolysis for their survival and proliferation [4].

Nicotinamide adenine dinucleotide (NAD^+^) is the main co-factor associated with cellular energetics and mediates redox reactions in crucial metabolic pathways, such as glycolysis, the TCA cycle and oxidative phosphorylation [5]. Additionally, it serves as a substrate for several NAD-degrading enzymes, including poly ADP-ribose polymerases (PARPs), mono (ADP-ribose) transferases (ARTs), NAD-ases/ADP ribosyl cyclases/cyclic ADP-ribose hydrolases (CD38/CD157) and sirtuins (SIRT). Subsequently, through these enzymes, NAD^+^ is involved in numerous biological processes, such as cellular proliferation, signaling and adhesion, calcium mobilization, cell cycle control, stress response, DNA damage, genomic integrity and transcriptional regulation [6]. NAD^+^ results from different molecular pathways, de novo from tryptophan, from nicotinic acid/nicotinic acid riboside (NA/NAR) through the Preiss–Handler pathway or through salvage pathways from nicotinamide/nicotinamide riboside (NAM/NR) and reduced nicotinamide riboside (NRH) [7,8]. In mammalian cells, NAM/NA are the most commonly used precursors to synthesize NAD^+^. Nicotinamide phosphoribosyltransferase (NAMPT) is the rate-limiting enzyme that catalyzes the phosphoribosylation of NAM to produce nicotinamide mononucleotide (NMN) [9,10]. Then, the latter is converted to NAD^+^, a process catalyzed by NMN adenyltransferase. Furthermore, NAMPT expression is upregulated in most cancer cells [11,12,13] and is associated with tumor progression [14,15]. Thus, targeting NAMPT has emerged as a promising strategy to eliminate malignant cells selectively, as they highly rely on NAD^+^ synthesis [16,17]. The first NAMPT inhibitor, **FK866**, also known as APO866 (Figure 1), was described by Hasmann et al. [18]. Its high killing ability on multiple cancer cell lines was observed in several preclinical studies [19,20,21,22,23]. This observation has provided a rationale for testing **FK866** (APO866) and another NAMPT inhibitor **CHS-828** (also known as GMX-1778, Figure 1) in clinical trials on hematological and solid malignancies (NCT00435084, NCT00432107, NCT00431912, NCT00457574 and NCT00724841). Despite their high anticancer activity observed in several preclinical studies, these inhibitors did not achieve the same efficacy in clinical trials. The clinical results showed a few dose-limiting toxicities and no significant tumor response [24,25,26]. The orally available NAMPT inhibitors **KPT-9274** and **OT-82** (Figure 1) have been tested as potential drugs in patients with solid tumors or with relapsed/refractory lymphoma [27,28]. Up to now, none of the published NAMPT inhibitors have become an anticancer agent. Thus, there is still a strong need for developing new potent and highly efficient NAMPT inhibitors.

We report here the synthesis of the three new NAMPT inhibitors: **JJ08**, **FEI191** and **FEI199**. We test them for their potential antitumor activities toward hematological malignancies. We show that these compounds are potent NAMPT inhibitors that profoundly deplete NAD(H) and NADP(H) after 24 h of incubation, which is followed by a strong, time-dependent increase in ROS production including cytosolic/mitochondrial superoxide anions and hydrogen peroxide. That increase correlates with ATP depletion and mitochondrial depolarization. We provide evidence that **JJ08**, **FEI191** and **FEI199** exhibit cell death at low nanomolar concentrations towards several hematopoietic malignant cells. Treatment of mouse xenografts with the three new NAMPT inhibitors significantly prolonged mouse survival. **JJ08** presented the most promising results as it abolished tumor growth completely.

## 2. Results

### 2.1. Synthesis of New Potent NAMPT Inhibitors

Within the European Health 7th framework project PANACREAS (grant agreement ID: 256986) we have synthesized a series of novel **FK866** (APO866) analogues and shown that some of them are very toxic toward pancreatic cell lines [29]. Applying an analogous synthetic route to that reported for the synthesis of **FK866** [29], we have prepared ((*E*)-*N*-(4-(1-(furan-2-carbonyl)piperidin-4-yl)butyl)-3-(pyridin-3-yl)acrylamide) (**JJ08**) which exchanges the benzamide moiety of **FK866** for a furan-2-carboxamide group. The synthesis starts with the reduction of the HCl salt of 4-(piperidin-4-yl)butyric acid (**1**) into alcohol **2**. The latter reacted with 2-furanoyl chloride and triethylamine, giving the corresponding carboxamide **3**. A Mitsunobu displacement of the alcoholic moiety of **3** with phathalimide gave **4**. Selective liberation of the primary amine **5** was realized by treatment with hydrazine chlorhydrate in ethanol. Amide coupling of **5** with (*E*)-3-(pyridin-3-yl) provided **JJ08** (Figure 1).

**FEI191** ((*Z*)-2-cyano-1-(5-(1-(furan-2-carbonyl)piperidin-4-yl)pentyl)-3-phenylguanidine) and **FEI199** ((*Z*)-2-cyano-1-((*E*)-5-(1-(furan-2-carbonyl)piperidin-4-yl)pent-2-en-1-yl)-3-phenylguanidine) are compounds in which the (pyridine-3-yl)acrylamide moiety of **JJ08** has been exchanged for a (pyridin-4-yl)cyanoguanidine moiety and the C4-tether for C5-tethers. These compounds can be seen as chimeric derivatives of **CHS-828**. The synthesis of **FEI191** (Figure 2) starts with the commercially available tert-butyl 4-(3-hydroxypropyl)piperidine-1-carboxylate (**6**), a primary alcohol that undergoes Dess–Martin oxidation into the corresponding aldehyde **7**. Wittig–Horner–Emmons olefination of **7** furnished the (*E*)-ene-ester **8** that was reduced into the corresponding (*E*)-allylic alcohol **9** by di(isobutyl)aluminum hydride in CH_2_Cl_2_. Catalytic hydrogenation of **8** provided ester **10**, which was reduced into the corresponding alcohol **11**. Mitsunobu displacement of the primary alcohol **11** with phthalimide gave **12**. The piperidine moiety of **12** was deprotected selectively on treatment with aqueous HCl in dioxane furnishing chlorhydrate **13**, which was converted into carboxamide **14** with 2-furoic chloride and trimethylamine. Selective liberation of the primary amine with hydrazine gave **15**, which was treated with phenyl (*Z*)-*N*′-cyano-*N*-(pyridin-4-yl)carbamimidate to provide **FEI191**.

The synthesis of **FEI199** starts with allylic alcohol **9** obtained above (Figure 2). Over four steps, and without purification of the intermediate products, crude **16** was obtained in 55% yield. The preparations followed the same procedures as those for the conversion of **11** into **15**. Treatment of **16** with phenyl (*Z*)-*N*′-cyano-*N*-(pyridin-4-yl)carbamimidate provided **FEI199** (Figure 3).

The detailed synthesis and characterization of **JJ08**, **FEI191** and **FEI199** can be found in Appendix A, p. 3–13.

### 2.2. JJ08, FEI191 and FEI199 Are Potent NAMPT Inhibitors That Efficiently Deplete Intracellular NAD^+^ Content

First, we assessed whether the novel compounds are indeed NAMPT inhibitors by examining their capacity to inhibit in vitro NAMPT activity. Using **FK866** (APO866), a prototype of NAMPT inhibitors, as a positive control, **JJ08**, **FEI191** and **FEI199** were tested in an NAMPT enzymatic inhibition assay. Figure 2 indicates that they were all potent NAMPT inhibitors, showing full inhibition of NAMPT. The direct consequence of NAMPT inhibition is the decrease in intracellular NAD^+^ content. Hence, we investigated whether treatment of hematopoietic malignant cells with the selected NAMPT inhibitors led to NAD^+^ depletion. To this end, we performed a time course analysis of intracellular NAD^+^ levels on four hematological cancer cell lines, including ML2, Jurkat, Namalwa and RPMI8226, which were treated with the selected compounds. As reported in Figure 3A–D, all tested NAMPT inhibitors fully depleted the NAD^+^ cell content within the first 24 h after treatment. Notably, in ML2 cells, an additional 8 h time point was recorded, indicating a fast drop in NAD^+^ levels (Figure 3A).

Taken together, these results indicate that all tested compounds are potent NAMPT inhibitors.

### 2.3. JJ08, FEI191 and FEI199 Induce Different Types of Cell Death in Several Hematological Malignancies in NAD^+^ Dependent Manner

NAD^+^ depletion has been proposed as a promising strategy to eliminate hematological malignancies [19,23,28,30,31]. We measured the cytotoxic activities of **JJ08**, **FEI191** and **FEI199** in the four aforementioned hematological cancer cell lines. The cell growth inhibitory effects were compared to that of **FK866** (APO866). As summarized in Table 1, the half-maximal inhibitory concentration (IC_50_) values of the tested inhibitors were in the low nanomolar range. Among them, **FEI199** was the most potent, with a measured IC_50_ that was lower than 0.3 nM in all malignancies.

To assess whether apoptosis is involved in the NAMPT inhibitor-induced cytotoxicity, malignant cells were first treated with the compounds for 96 h and subsequently double stained with ANXN/7AAD and analyzed by flow cytometry. As shown for ML2 cells and additional different malignant hematopoietic cell lines, all the inhibitors induced early (ANXN+/7AAD-) and late apoptotic (7AAD+) cell death at drug concentrations ranging between 0.1 and 10 nM. **FEI199** induced maximal cell killings at very low concentrations (≤0.5 nM) on all tested cell lines, whereas at similar concentrations, **FK866** (APO866) and **JJ08** induced cell death at only between 20 and 75%, depending on the cell line (Figure 4A–D). Moreover, **FEI191** and **FEI199** induced more late apoptotic/necrotic (7AAD+) than early apoptotic cell death (ANXN+/7AAD-) compared with APO866. To provide additional evidence of the involvement of apoptosis in the cell death induced by the new NAMPT inhibitors, we assessed the activation of various caspases, including CASP-3, CASP-8 and CASP-9. Hematopoietic malignant cells were treated for 72 h with the compounds and caspase activation was assessed using the specific CaspGLOW™ Red Active probes specific for each caspase and flow cytometry. The results show a strong increase in CASP-3, CASP-8 and CASP-9 activities in treated versus untreated cells (Figure 5), suggesting that caspase-dependent apoptosis contributes to the antitumor activity of the tested compounds. 

Another possible type of cell death is necrosis, which correlates with the release of the cytosolic enzymes, especially lactate dehydrogenase (LDH), into the extracellular space. Therefore, the detection of LDH in the medium is used as a marker of necrotic cell death [32]. Accordingly, we monitored necrotic cell death in time-dependent analyses, as well as the drug effect on cell proliferation in ML2 and Jurkat cells cultured with or without the new NAMPT inhibitors. As shown in Figure 6A,B, LDH release in the medium significantly increased over time in leukemic cells treated with the NAMPT inhibitors compared to untreated ones, whereas cell proliferation decreased over time and only approximately 40% of proliferating cells remained at 48 h after treatment (Figure 6C,D). These finding show the involvement of necrotic cell death in treatment with NAMPT inhibitors.

To demonstrate that the antitumor activity of the new NAMPT inhibitors was due to NAD^+^ depletion, we evaluated the ability of NAM and NA (precursors involved in the NAD^+^ biosynthesis), as well as NAD^+^, to abrogate the cell death caused by our compounds. Extracellular supplementation in excess with NAD^+^ or its precursors fully restored the viability of the cells despite the presence of the inhibitors (Figure 7). Interestingly, the supplementation with NA (but not with NAM or NAD^+^) did not protect Namalwa cells from cell death in response to treatment with the NAMPT inhibitors (Figure 7C). This can be explained by the fact that Namalwa cells have a naturally very low expression of the nicotinic acid phosphoribosyltransferase (NAPRT) gene [33], which is required to utilize NA in NAD^+^ biosynthesis.

Collectively, these results indicate that the new NAMPT inhibitors induce both apoptotic and necrotic cell death in an NAD^+^-dependent manner in several human hematopoietic malignant cells.

### 2.4. Treatment with JJ08, FEI191 and FEI199 Induces High Levels of ROS Production and ATP Depletion in Hematological Malignant Cells

The first consequence of NAMPT inhibition is NAD^+^ depletion, which occurs within 24 h and will subsequently result in a profound decrease in NADP(H). To verify this hypothesis, we evaluated the intracellular NADP(H) content in myeloid leukemia cells upon treatment with NAMPT inhibitors. As shown in Figure 8, treatment with the compounds significantly depleted NADP(H) cell content compared to untreated cells. Since NADPH, a powerful cell antioxidant, is directly involved in redox reactions and is essential to maintain cellular homeostasis, its depletion is expected to generate high levels of oxidative stress. Therefore, cytosolic and mitochondrial superoxide anions, as well as intracellular hydrogen peroxide, were measured in hematopoietic malignant cells treated with the new compounds, using DHE, MitoSOX and carboxy-H2DCFDA probes, respectively. In accordance with our hypothesis, Figure 9 shows that the new NAMPT inhibitors increased the levels of various ROS in all treated cell types. High ROS production is known to be detrimental for the cells, since it oxidizes proteins, lipids and cell organelles, including mitochondria, resulting in ATP depletion [19,34] and ultimately leading to cell death [35]. As expected, the treatment of hematopoietic malignant cells with the new NAMPT inhibitors led to an ATP loss in a time-dependent manner (Figure 10A), which was followed by mitochondrial membrane depolarization (Figure 10B), that ultimately resulted in cell death at 96 h (Figure 10C). To provide strong evidence that high levels of ROS production are the main driver of these events that led to cell death, we monitored cell death in hematopoietic malignant cells treated with these compounds in the presence or absence of catalase, a potent H_2_O_2_ scavenger [36,37]. As shown in Figure 11, the supplementation with catalase did not prevent NAD^+^ depletion in ML2 cells (Figure 11A), but it fully abrogated the loss in ATP (Figure 11B) and MMP (Figure 11C), as well as the ultimate cell death (Figure 11D), in response to all of the tested NAMPT inhibitors. Moreover, the supplementation with catalase also prevented the cell death caused by NAMPT inhibitors at 72 h in Jurkat and RPMI8226 cell lines (Figure 11E–G).

Collectively, these results indicate that all of the tested NAMPT inhibitors significantly depleted cellular NADP(H) content, resulting in a burst of ROS production. In turn, this induces the loss of ATP, which is followed by mitochondrial membrane depolarization and ultimately leads to cell death. Importantly, oxidative stress appears to be the main cause of cancer cell death after NAMPT inhibitor treatment.

### 2.5. The Therapeutic Activity of JJ088 in SCID Mice Bearing Burkitt Lymphoma Is Superior to That of FEI191 and FEI199

The promising results presented above led us to explore the potential therapeutic efficacy of the new NAMPT inhibitors in a mouse xenograft model of Burkitt lymphoma. To this end, NAMPT inhibitors (10 mg/kg) were administered intraperitoneally (I.P.) to mice with established Namalwa tumors (human Burkitt lymphoma cell line) and tumor growth was monitored over time. As shown in Figure 12, treatment with the new NAMPT inhibitors exerted a significant therapeutic effect (Figure 12A) and significantly prolonged overall mice survival compared to untreated control animals (Figure 12B, log-rank test, P < 0.05). Interestingly, treatment with **JJ08** completely eradicated tumor growth 5 days after administration. Instead, **FEI191** and **FEI199** did not stop tumor progression, but significantly delayed it compared to the vehicle-injected group (Figure 12A), suggesting that the in vitro efficiencies of **FEI191** and **FEI199** do not translate into equally potent in vivo activities. To understand why **FEI191** and **FEI199** were less efficient in abrogating tumor growth than **JJ08**, pharmacokinetics (PK) studies were carried out. Plasma concentrations of the compounds were monitored in mice (n=3) after I.P. administration for up to 24 h (Appendix A). Then, PK parameters were derived and are presented in Table 2. The PK values of **FK866** (APO866), which is known to effectively abrogate tumor growth in vivo, are shown as a reference [19]. The measured compound concentrations used for the PK data analysis are given in Appendix A. Notably, the concentrations of **FEI199** measured after 8 h and 24 h were excluded, as the analytical responses were below the lower limit of quantification of the method. For **FEI191**, concentrations measured at 8 h and 24 h were also excluded from the PK data analysis due to carryover issues.

The four compounds showed broadly similar PK profiles, but **FK866** (APO866) distinctively appeared to provide the best systemic exposure (i.e., AUC0-24). In addition, the plasma concentrations of APO866 were less variable between the mice samples (Appendix A). The maximum plasma concentrations (C_max_) of **JJ08** and APO866 were comparable, while the AUC_0-24_ of **JJ08** appeared to be approximately 2-fold lower than that of APO866. This difference is mainly due to the early time points, which contribute significantly to the calculated AUC_0-24_ values. Similarly, the calculated apparent drug clearance (CL/F) for **JJ08** was found to be 2-fold higher, whereas its half-life (T_1/2_) appeared to be two times shorter as compared to APO866, indicating that **JJ08** was cleared faster from circulation. Finally, based on the calculated apparent terminal volume of distribution (Vz/F), both **JJ08** and APO866 compounds seemed to be well absorbed into tissues and/or highly metabolized.

Regarding **FEI191** and **FEI199**, the terminal rate constant, λz, value (and therefore, T_1/2_ and Vz/F) could not be assessed because their terminal phases were insufficiently characterized due to analytical issues, as discussed above. However, the maximal concentrations (C_max_) were lower than those of APO866 or **JJ08** and the clearance of **FEI191** and **FEI199** proved to be more than 4-fold higher, suggesting faster drug elimination.

Taken together, our in vivo data indicate that the new NAMPT inhibitors delayed and/or prevented tumor growth in a mouse Burkitt lymphoma model, with **JJ08** being the most potent anticancer agent. Furthermore, **JJ08** had very similar PK parameters to APO866, whereas both **FEI** compounds exhibited lower concentrations in the blood after administration, which could explain their lower anticancer activities in vivo.

## 3. Discussion

In this study, we report the synthesis and evaluation of the therapeutic efficacies of three novel NAMPT inhibitors, **JJ08**, **FEI191** and **FEI199**, in hematological malignancies. We show that the new compounds have a broad antitumor activity against various hematological malignancies. In agreement with our previous studies on APO866, a prototype NAMPT inhibitor [19,36,38,39], we found that the new NAMPT inhibitors are highly toxic towards leukemia (AML and ALL), lymphoma (Burkitt) and multiple myeloma (MM) cells. Mechanistically, these compounds caused a strong NAD^+^ depletion that led to exhaustion of NADPH, which in turn resulted in a burst of oxidative stress. The high levels of ROS induced by these compounds disrupted the mitochondrial membrane integrity, causing ATP depletion and cell death. Scavenging ROS production with catalase abrogated cell death induced by NAMPT inhibitors, despite NAD^+^ depletion, pointing out the major contribution of oxidative stress to the antitumor activity of APO866 and of the new NAMPT inhibitors. In addition, we demonstrated that the new NAMPT inhibitors induced different types of cell death, including both caspase-dependent and caspase-independent apoptosis, but also necrotic cell death. Therefore, their mode of action described in this study is similar to that reported previously for NAMPT inhibitors [19,30,36,38,39,40,41],indicating that although these compounds have different chemical structures, they have common mechanisms involved with cell death.

Importantly, in vivo administration of the new NAMPT inhibitors as a single agent prevented and/or delayed tumor growth in an animal model of human Burkitt lymphoma and significantly prolonged median survival, thereby underlining the therapeutic potential of these molecules. It is noteworthy to mention that **JJ08** fully eradicated tumor growth and allowed mouse disease-free survival. In line with the in vivo data, **JJ08**, as well as APO866, exhibited the best PK properties when compared to those of both **FEI** compounds.

The search for new NAMPT inhibitors is motivated by the need to identify novel drugs that counter cancer progression and thereby increase patient life expectancy and quality of life, a goal of a high priority. In this endeavor, the development of anticancer therapies targeting the NAMPT-mediated NAD^+^ biosynthetic pathway represents a promising strategy and should have broad clinical implications. We and others have demonstrated that NAMPT inhibitors exhibit high efficacy against a wide range of human solid tumors and blood cancers, without significant toxicity to laboratory animal models [19,29,36,42,43,44,45,46,47,48,49]. In an effort to discover new anticancer agents, we here have identified three novel NAMPT inhibitors with broad and strong anti-leukemic/lymphoma activity. Among them, **JJ08** exhibited a promising profile as an overall potent antitumor agent both in vitro and in vivo, despite the fact that in vitro, **FEI191** and **FEI199** had higher antitumor activity than **JJ08**. This discrepancy between in vitro and in vivo studies is most probably related to the worse PK profiles of **FEI** compounds. Indeed, PK analyses showed that **FEI** compounds were rapidly cleared out of body circulation compared to **JJ08** (or APO866). Moreover, the calculated clearance of **FEI** compounds was at least 4-fold higher, the C_max_ was lower and their AUCs were approximately 2-to-9-fold smaller than those of **JJ08** (or APO866). The apparent volume of distribution calculated for all molecules indicated that they are highly absorbed into the tissues and/or highly metabolized. The calculated clearance for all compounds indirectly suggested that these novel NAMPT inhibitors are molecules with high hepatic excretion. Further studies aiming at improving the PK properties of novel NAMPT inhibitors are needed.

To put our results in a global context, one should keep in mind that the striking antitumor activity of NAMPT inhibitors reported in several studies is closely correlated with their in vitro experimental conditions. For instance, RPMI medium widely used for cell culture contains only nicotinamide as an NAD^+^ precursor. In our study, and in many preclinical studies, the major (if not the only) source of NAD^+^ synthesis was also nicotinamide, indicating that only one route of NAD^+^ synthesis, namely, the salvage pathway, is activated within these experimental settings. In a real life situation, where many NAD^+^ precursors could be present in a tumor environment, blocking only one pathway of NAD^+^ synthesis would not be sufficient and this could greatly contribute to the loss of the therapeutic efficacy of NAMPT inhibitors. In agreement with this scenario, we and others have demonstrated that the levels and/or presence of NAD^+^ precursors (other than nicotinamide) considerably affect the antitumor efficiency of NAMPT inhibitors [39,50]. The loss of the efficacy of NAMPT inhibitors in the latter circumstance was mainly due to the activation of the alternative NAD^+^ production pathways. We also showed that gut microbiota played a crucial role in host NAD^+^ metabolism, as they contribute to resistance to NAMPT inhibitors [39]. These observations should be taken into consideration in future clinical trials, for instance, the nature and level of NAD^+^ precursors or alternatively targeting more than one route of NAD^+^ synthesis should be investigated.

In this study, we showed that the novel NAMPT inhibitors delayed or eradicated the tumor growth and thus significantly prolonged xenografted mouse survival, without evident signs of toxicity including loss of body weight, lethargy, rough coat or premature death. However, in clinical trials, the common dose-limiting toxicities were thrombocytopenia and a variety of gastrointestinal symptoms [24,25,26,51]. Therefore, the strategies to limit off-target toxicities need to be refined. **FEI191** and **FEI199** had high activities in vitro. Therapeutic modalities to significantly boost their in vivo activities and reduce their systemic associated toxicities should be explored. In this line, the next generation of NAMPT inhibitors can be conjugated to antibodies (creating antibody–drug conjugates, or ADCs). In this drug delivery system, the inhibitor is conjugated to the antibody that targets the antigens/proteins specifically expressed in cancer cells, thus allowing specific inhibitor delivery. Using such a strategy, several investigators [52,53,54] have demonstrated the antitumor efficacy of ADCs with NAMPT inhibitors in different mouse xenograft models. Only mild, reversible hematologic side effects were observed with ADCs in toxicological in vivo studies, with no signs of retinal or cardiac toxicities, as reported for NAMPT inhibitors alone in preclinical studies [52]. These findings open a new era in clinical trials to specifically target and improve the therapeutic window of NAMPT inhibition.

## 4. Conclusions

In summary, we have synthesized three novel NAMPT inhibitors: **JJ08**, **FEI191** and **FEI199**. They are strong growth inhibitors of cancer cells from numerous hematological malignancies. Our in vitro and in vivo data demonstrate that these compounds are potent anticancer agents. **JJ08** shows the best efficacy and is well tolerated in the mouse xenograft model of Burkitt lymphoma. We propose that **JJ08** should undergo further clinical development for the treatment of hematologic malignancies.

## 5. Materials and Methods

### 5.1. Cell Lines and Culture Conditions

Four hematological cell lines (ML2—acute myeloid leukemia; Jurkat—acute lymphoblastic leukemia; Namalwa—Burkitt lymphoma; and RPMI8226—multiple myeloma) were purchased from DSMZ (German Collection of Microorganisms and Cell Cultures, Braunschweig, Germany) or ATCC. 

All cells were cultured in RPMI medium (Invitrogen AG, 61870-01) supplemented with 10% heat inactivated fetal calf serum (Amimed, 2-01F30-I) and 1% penicillin/streptomycin at 37 °C (Amimed, 4-01F00-H) in a humidified atmosphere of 95% air and 5% CO_2_.

### 5.2. NAMPT Enzymatic Activity Assay

The ability of **FK866** (APO866) analogues to inhibit NAMPT activity was assessed with an NAMPT Activity Assay Kit (Colorimetric) (Abcam, ab221819, Cambridge, UK) according to the manufacturer’s instructions. Briefly, NAMPT inhibitors were dissolved in DMSO to a final concentration of 1 µM and distributed in a 96-well plate in duplicate. Then, a reaction mix containing assay buffer, ATP, NMNAT1, NAM, PRPP and ddH_2_O was added and the plate was incubated at 30 °C for 60 min. After, to measure the generated NAD^+^, a mix of WST-1, ADH, diaphorase and ethanol was added to the wells. The absorbance was measured in kinetic mode at 450 nm on a microplate reader for 45 min at 30 °C. 

### 5.3. Flow Cytometry Analyses

The cellular effects of **FK866** (APO866) and the new NAMPT inhibitors, **JJ08**, **FEI191** and **FEI199**, on hematopoietic malignant cells were evaluated using a Beckman Coulter Cytomics Gallios flow cytometer (Beckman Coulter International S.A., Nyon, Switzerland). The measured parameters included cell death, mitochondrial membrane potential (MMP), reactive oxygen species (ROS) production and caspase activation.

### 5.4. Characterization of Cell Death

To determine the cell death induced by NAMPT inhibitors, malignant cells were stained with ANNEXIN-V (ANXN, eBioscience, BMS306FI/300) and 7-aminoactinomycin D (7AAD, Immunotech, A07704) as described by the manufacturer and analyzed using flow cytometry. Dead cells were identified as ANXN+7AAD+ /7AAD+ and early apoptotic cells as ANXN+ 7AAD-. Specific cell death induced by inhibitors was calculated using the following formula: percent of cell death induced by compound = [(S – C) / (100 – C)] × 100; where S = treated sample cell death and C = untreated sample cell death.

### 5.5. Analysis of Mitochondrial Membrane Potential

Mitochondrial membrane depolarization was determined using tetramethylrhodamine methyl ester (TMRM, ThermoFisher Scientific, T668) according to the manufacturer’s protocol. TMRM is a cationic, cell-permeant, red-orange fluorescent dye that accumulates in polarized mitochondria, but it is released after their depolarization. Untreated or treated cells were harvested, centrifuged and resuspended in culture medium containing 50 nM TMRM, and then incubated at 37 °C for 30 min in the dark. Cells were washed twice with PBS and immediately analyzed using flow cytometry.

### 5.6. Detection of Cellular and Mitochondrial Reactive Oxygen Species (ROS)

Various types of ROS were determined in untreated and drug-treated hematopoietic malignant cells by flow cytometry using live-cell permeant specific fluorogenic probes. Dihydroethidium (DHE, Marker Gene Technologies, M1241) was used as probe for detection of the cytosolic superoxide anion (cO2•-), MitoSox (Molecular Probes, M36008) was used as probe for detection of the mitochondrial superoxide anion (mO2•-) and 6-carboxy-2,7-dichlorodihydrofluorescein diacetate (carboxy-H2DCFDA; Molecular Probes, C-400) was used as probe for detection of H_2_O_2_. DHE was oxidized to red fluorescent ethidium by cytosolic superoxide and MitoSOX was selectively targeted to mitochondria, where it was oxidized by superoxide and exhibited red fluorescence. Carboxy-H2DCFDA was cleaved by esterase to yield DCFH, a polar nonfluorescent product, but in the presence of hydrogen peroxide, the latter is oxidized to a green fluorescent product, dichlorofluorescent (DCF). For cell staining, cells were centrifuged and the pellets were resuspended in PBS with a final concentration of 5 μM for each probe. The mixture was incubated in the dark at 37 °C for 15 min. Then, the cell suspension was analyzed using flow cytometry within 20 min.

### 5.7. Detection of Caspases Activation

Activation of various caspases was assessed using flow cytometry and specific CaspGLOW™ Red Active (BioVision, K190, Cambridge, UK) for following caspases: CASP3 (BioVision Inc., BV-K193-100), CASPASE 8 (CASP8; BioVision Inc., BV-K198-100) and CASPASE 9 (CASP9; BioVision Inc., BV-K199-25). The CaspGLOW assays offer a convenient way for measuring activated caspases in living cells. The assay uses a specific inhibitor for each caspase conjugated to sulforhodamine as a fluorescent marker, which is cell permeable, nontoxic and irreversibly binds in specific manner to activated caspase in apoptotic cells. The red fluorescence label allows for direct detection of activated caspase in apoptotic cells by flow cytometry. Cell staining was performed according to the manufacturer’s information and analyzed.

### 5.8. Quantification of Intracellular NAD^+^, NADP(H) and ATP Contents

Cells (1 × 10^6^/mL) in the log growth phase were seeded in a 6-well plate in the presence or absence of the NAMPT inhibitors. At each time point, 800 µL of cells was centrifuged at 900 g (2000 rpm) for 5 min and washed with cold PBS. Then, the supernatant was discarded and cells were resuspended in 300 µL of lysis buffer (20 mM NaHCO_3_, 100 mM Na_2_CO_3_) and kept at –80 °C for at least 4 h before analysis.

Total NAD^+^ content was measured in cell lysates using a biochemical assay described previously [18]. Briefly, cell lysates (20 µL) were plated in a 96-well flat bottom plate. A standard curve was generated using a 1:3 serial dilution in lysis buffer of a β-NAD^+^ stock solution. Cycling buffer (160 µL) was added into each well and the plate was incubated for 5 min at 37 °C. Afterwards, ethanol (20 µL), pre-warmed to 37 °C, was added into each well and the plate was incubated for an additional 5 min at 37 °C. The absorbance was measured in kinetic mode at 570 nm after 5, 10, 15, 20 and 30 min at 37 °C on a spectrophotometer. The amount of NAD^+^ in each sample was normalized to the protein content for each test sample at each time point.

The NADP^+^ and NADPH contents in the cells were determined separately using an NADP/NADPH-GloTM kit (Promega, G9081, Madison, WI, USA) and according to the manufacturer’s protocol.

The total ATP cell content was quantified using an ATP determination Kit (Life Technologies, A22066, Carlsbad, CA, USA) according to the manufacturer’s instructions.

### 5.9. Detection of Necrotic Cell Death with LDH Assay

The LDH release quantification was performed using a colorimetric CyQUANT LDH Cytotoxicity Assay (Invitrogen, C20300, Carlsbad, CA, USA). Lactate dehydrogenase (LDH) is a cytosolic enzyme that is released into the cell culture medium upon the disruption of the plasma membrane, indicating the necrotic type of death. LDH is quantified in the media in enzymatic reactions. Firstly, LDH catalyzes the conversion of lactate to pyruvate with the accompanying reduction of NAD^+^ to NADH. Then, the added diaphorase oxidizes NADH, which leads to the reduction of a tetrazolium salt to a red formazan. The amount of formulated formazan is directly proportional to the total LDH released into the media. Here, cells (1 × 105/mL) in the log growth phase were seeded in a 24-well plate in the presence or absence of NAMPT inhibitors. At each time point, 100 µL of cells was transferred to a 96-well plate and the reaction mixture from the kit was added. The plate was then incubated at RT for 30 min and protected from light. Afterwards, the stop solution was added and the absorbance was measured at 490 nm with a spectrophotometer. The higher the absorbance intensity in the sample, the more LDH is released to the culture medium.

### 5.10. Cell Proliferation Determination

The cell proliferation was assessed with alamarBlue® reagent (Bio-Rad, BUF012B, Hercules, CA, USA), which is based on REDOX reaction by viable cells. Specifically, resazurin sodium salt is reduced by the reducing environment of metabolically active cells to the highly fluorescence resorufin sodium salt. Cells were seeded in a 24-well plate (1 × 105/mL) and treated with NAMPT inhibitors. After incubation, at each time point, 200 µL of cells was transferred to a 96-well plate and alamarBlue® dye (20 µL) was added, then the plate was incubated for 4 h in 37 °C in the dark. At the end, the absorbance at 570 and 600 nm was measured. Proliferation is depicted as a percentage of the control.

### 5.11. Therapeutic Efficacy Evaluation of Novel NAMPT Inhibitors Using a Mouse Xenograft Model of Human Burkitt Lymphoma

The new NAMPT inhibitors (**JJ08**, **FEI191** and **FEI199** (in comparison with lead compound, **FK866** (APO866))) were evaluated in vivo in a mouse xenograft model of human Burkitt lymphoma. Twenty non-leaky C.B.-17 SCID mice (8 to 10 weeks old; Iffa Credo, L’Arbresle, France) were housed in micro-isolator cages in a specific pathogen-free room in the animal facility at the University Hospital of Lausanne. Firstly, the mice spent one week alone to acclimatize to their new environment. All animals were handled according to the institutional regulations and with the prior approval of the animal ethic committee of the University of Lausanne. Manipulations were performed in sterile conditions under a laminar flow hood. Firstly, Namalwa cells (1 × 10^7^) were injected subcutaneously into the mouse flank side. Once the tumors became palpable and reached a size between 100 and 150 mm^3^, mice (n = five/ group) were randomized into control and treated groups. The drugs were administered intraperitoneally (10 mg/kg body weight) in 200 µL 0.9% saline twice a day for 4 days, repeated weekly over 3 weeks. The control group was treated only with 200 µL 0.9% saline. Every day, the animals were monitored for any signs of illness, and in cases where the tumor size reached a diameter of 15 mm, they were sacrificed immediately.

### 5.12. Analytical Method of Pharmacokinetic Studies In Vivo

Concentration measurements in mice EDTA plasma samples were performed using a Vanquish Flex ultra-high-performance liquid chromatography (UHPLC) system attached to a TSQ Quantis^TM^ triple quadrupole mass spectrometer (MS) (ThermoFisher Scientific, Waltham, MA, USA). The chromatographic column was a Luna Omega Polar C18 3 µm, 50 × 2.1 mm from Phenomenex (Torrance, CA, USA), kept at 40 °C in a UHPLC oven. The mobile phase was made of water and acetonitrile (ACN) with 0.1% formic acid in each. The gradient program ranged from 20 to 95% ACN in 1.5 min and the total method duration (including equilibration for the next injection) was 3 min. The flow rate and injection volume were 0.5 mL/min and 5 µL, respectively.

For the sample preparation, 90 µL of ACN was added to an aliquot of 30 µL of mouse plasma for protein precipitation. The mixture was then centrifugated at 14,000 rpm and the supernatant was directly injected in the UHPLC–MS.

### 5.13. Pharmacokinetic Analyses

Drug plasma concentrations were measured at selected time points after intraperitoneal administration in mice (sacrificed mice in triplicates for each time point). Samples were analyzed on two separate occasions for each sampling. Then, pharmacokinetic (PK) parameters were computed using standard non-compartmental calculations for geometric means of the measured concentrations using the “PKNCA R Package” (R version 4.0.2, R Development Core Team, http://www.r-project.org/ access date: 4 February 2023). 

The area under the curve (AUC0-24) was calculated for the four drugs using the trapezoidal and log-trapezoidal rule when appropriate. The terminal rate constant (λz) was approximated using the slope of the terminal phase, while the half-life (T_1/2_) was calculated as ln(2)/λz, the apparent clearance (CL/F) as the dose divided by AUC0-24 and the apparent volume of distribution (Vz/F) as (CL/F)/λz.

### 5.14. Statistical Analysis

All experiments were performed in triplicate and data are expressed as means with the standard error of the mean (SEM), unless otherwise noted. Unpaired t-tests were performed to test differences in pre- and post-treatment malignant cells. The Kaplan–Meier survival method using a long rank test was applied for the analyses of animal survival studies. GraphPad Prism version 9.1.0 (GraphPad Software, San Diego, CA, USA) was used for statistical analysis. *p* values less than 0.05 were considered statistically significant.

## Data Availability

The data presented in this study are available on request from the corresponding author.

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
