# Peer review of "Anticancer Activities of Novel Nicotinamide Phosphoribosyltransferase Inhibitors in Hematological Malignancies"

_molecules, 2023, doi:10.3390/molecules28041897_

Round 1

Reviewer 1 Report

The article entitled "Anticancer Activities of Novel Nicotinamide Phosphoribosyl-transferase Inhibitors in Haematological Malignancies" describes the synthesis and anticancer activity of new NAMPT inhibitors against several haematopoietic malignant cells. Their effect in in vitro and in vivo models are reported, as well as the mechanism of action, due to the depletion of NAD+ and NADP(H) and time-dependent reactive oxygen species (ROS) production.

The article is well written and the results are interesting for the field, however some corrections about data organization and presentation need to be made to improve the quality of the manuscript. Hence, it is suitable for publication after the following major revisions:

-          Since authors report a series of novel NAMPT inhibitors, the synthetic schemes must be moved from the supporting informations to the results paragraph (2.1);

-          In Figure 1, please make a clear difference among authors’ compounds (in particular their role as APO866 and CHS-828 analogues) and those synthetised by other groups;

-          Some figures and all tables have poor resolution;

-          In paraghraph 2.3, add the cytotoxic evaluation in healthy cells;

-          Please check that in vitro and in vivo are always written in italics;

-          Conclusions are missing

Author Response

Reviewer 1

Comments and Suggestions for Authors

The article entitled "Anticancer Activities of Novel Nicotinamide Phosphoribosyl-transferase Inhibitors in Haematological Malignancies" describes the synthesis and anticancer activity of new NAMPT inhibitors against several haematopoietic malignant cells. Their effect in in vitro and in vivo models are reported, as well as the mechanism of action, due to the depletion of NAD+ and NADP(H) and time-dependent reactive oxygen species (ROS) production.

The article is well written and the results are interesting for the field, however some corrections about data organization and presentation need to be made to improve the quality of the manuscript. Hence, it is suitable for publication after the following major revisions:

-          Since authors report a series of novel NAMPT inhibitors, the synthetic schemes must be moved from the supporting informations to the results paragtionraph (2.1);

We thank the reviewer for this comment. We now provide in the results section, a new paragraph that includes “the synthetic schemes”- as well as details on the rationale, optimization and methods used to generate the new compounds. However, we kept in the supplementary file, the details on chemical synthesis.

-          In Figure 1, please make a clear difference among authors’ compounds (in particular their role as APO866 and CHS-828 analogues) and those synthetised by other groups;

The figure 1 is now adapted according to the reviewer 1’s comment. Furthermore, additional explanation of the differences between structures of the compounds are included in the figure 1 description.

-          Some figures and all tables have poor resolution;

We agree with the reviewer 1’s comment. We now provide all the figures with high resolution (600 dpi) and in good file format (TIF).

-          In paraghraph 2.3, add the cytotoxic evaluation in healthy cells;

We agree with the reviewer’s comment that the evaluation of the cytotoxicity of our compounds in healthy primary cells would have brought complementary evidence on their killing selectivity. However, the use of human sample requires an approval authorization of our institutional Review Board (Ethics Committee) that we do not have currently. This administrative procedure is cumbersome and time consuming. Nevertheless, based on our previous studies and those from other investigators on the cytotoxicity of NAMPT inhibitors, it is well established that latter compounds are not toxic to healthy cells (Nahimana et al, 2009 PMID: 19196867; Cea et al, 2012 PMID: 22955917; Hjarnaa et al, 1999 PMID: 10582695; Korotchkina et al, 2020 PMID: 31896781). In line with this assumption, the new NAMPT inhibitors exhibited antitumor activity in animal models of human hematologic malignancies without significant toxicity in animals. In terms of safety, no evident signs of toxicity were observed in mice treated with 10 mg/kg of the new molecules and they were well tolerated with no premature deaths.

-       Please check that in vitro and in vivo are always written in italics;

We thank the reviewer for this remark; all in vitro and in vivo are corrected accordingly.

-        Conclusions are missing

We thank the reviewer 1 for this comment. We have now added the Conclusion section in the revised manuscript.

Reviewer 2 Report

The authors report the synthesis of new NAMPT inhibitors, JJ08, FEI191 and FEI199 that exhibit a broad anticancer activity, and demonstrate efficient anticancer activity of the new NAMPT inhibitor JJ08 in nanomolar range on several haematopoietic malignant cells.

The topic is quite interesting and the manuscript is well written. The manuscript may be published after minor correction.

1.      all the figs are blurry.

2.      In line 563, 1X    107 should be  1X107,the same as in line 565 150 mm3 should be 150 mm3, the author should check all the  manuscript.

Author Response

RESPONSES TO THE COMMENTS FROM REVIEWERS

Reviewers’ Comments:

Reviewer 2

Comments and Suggestions for Authors

The authors report the synthesis of new NAMPT inhibitors, JJ08, FEI191 and FEI199 that exhibit a broad anticancer activity, and demonstrate efficient anticancer activity of the new NAMPT inhibitor JJ08 in nanomolar range on several haematopoietic malignant cells.

The topic is quite interesting and the manuscript is well written. The manuscript may be published after minor correction.

  1. all the figs are blurry.

We thank the reviewer for this remark and we now provide all figures in good format (TIF) with high resolution (600 dpi),

  1. In line 563, 1X 107 should be  1X107 ,the same as in line 565 150 mm3 should be 150 mm3, the author should check all the  manuscript.

We thank the reviewer for this remark. All these modifications (superscripts) are now included in the revised manuscript.

Reviewer 3 Report

This manuscript by Biniecka et al. describes “Anticancer Activities of Novel Nicotinamide Phosphoribosyl-transferase Inhibitors in Haematological Malignancies”. These compounds displayed potent anticancer activity against tested cancer cell lines below 1.0 nM. Authors have systematically explored in vitro, in vivo, and pharmacokinetics studies to explore these compounds. Introduction, results and discussion are written well in this manuscript but its hard to read data in figures 3, 4, 6, 7, 8, and 10-12. I would recommend authors to upload all figures again with clear x-axis, y-axis, and legends so that be correlated to text. Other comments are as below

1.     Abstract can be reduced up to 200 words and keywords related to this work can be included below abstract.

2.     These compounds displayed potent cytotoxicity against cancer cell lines, did authors test these against any normal cell line?

3.     Line 311, page 12, Was JJ-08 administered at dose of 30 mg/kg (light blue line) but in text it shows 10 mg/kg?

4.  The references should be as per journal format.

Author Response

RESPONSES TO THE COMMENTS FROM REVIEWERS

Reviewers’ Comments:

Reviewer 3

Comments and Suggestions for Authors

This manuscript by Biniecka et al. describes “Anticancer Activities of Novel Nicotinamide Phosphoribosyl-transferase Inhibitors in Haematological Malignancies”. These compounds displayed potent anticancer activity against tested cancer cell lines below 1.0 nM. Authors have systematically explored in vitro, in vivo, and pharmacokinetics studies to explore these compounds. Introduction, results and discussion are written well in this manuscript but its hard to read data in figures 3, 4, 6, 7, 8, and 10-12. I would recommend authors to upload all figures again with clear x-axis, y-axis, and legends so that be correlated to text. Other comments are as below

  1. Abstract can be reduced up to 200 words and keywords related to this work can be included below abstract.

As suggested by the reviewer 2, the abstract has been shortened to 200 words and the keywords are provided below the abstract.

  1. These compounds displayed potent cytotoxicity against cancer cell lines, did authors test these against any normal cell line?

We agree with the reviewer’s comment that the evaluation of the cytotoxicity of our compounds in healthy primary cells would have brought complementary evidence on their killing selectivity but the use of human sample requires an approval authorization of our institutional Review Board (Ethics Committee) that we do not have currently. This administrative procedure is cumbersome and time consuming. Nevertheless, based on our previous studies and those from other investigatorsm on the cytotoxicity of NAMPT inhibitors, it is well known that latter compounds are not toxic to healthy cells (Nahimana et al, 2009 PMID: 19196867; Cea et al, 2012 PMID: 22955917; Hjarnaa et al, 1999 PMID: 10582695; Korotchkina et al, 2020 PMID: 31896781). In line with this assumption, the new NAMPT inhibitors exhibited antitumor activity in animal models of human hematologic malignancies without significant toxicity in animals. In terms of safety, no evident signs of toxicity were observed in mice treated with 10 mg/kg of the new molecules and they were well tolerated with no premature deaths.

  1. Line 311, page 12, Was JJ-08 administered at dose of 30 mg/kg (light blue line) but in text it shows 10 mg/kg?

We thank the reviewer for pointing out this error. Actually, JJ08 was administered at dose of 10mg/kg and this modification is now included in the Figure 12 description.

  1. The references should be as per journal format

The references are now updated as per journal requirements.

Reviewer 4 Report

The authors have reported the synthesis  and evaluation of the antitumour activity of new Nicotinamide phosphoribosyltransferase (NAMPT) inhibitors (JJ08, FEI191 and FEI199) in haematological malignancies. This article is not suitable for publishing in the current form for the following reasons:

- How to optimize is not clear. What methods have been used for sample generating and what algorithms have been used. A complete section with details is necessary for this subject;

- Why the authors did not add keywords?!;

- Figures 3,4,6,7,8,10,11 and 12 are not clear and they are hard to read;

- Generally, there are mistakes in some sentences of the manuscript. So related corrections should be applied;  

- The References section has not been well organized.

Author Response

RESPONSES TO THE COMMENTS FROM REVIEWERS

Reviewers’ Comments:

Reviewer 4

Comments and Suggestions for Authors

The authors have reported the synthesis  and evaluation of the antitumour activity of new Nicotinamide phosphoribosyltransferase (NAMPT) inhibitors (JJ08, FEI191 and FEI199) in haematological malignancies. This article is not suitable for publishing in the current form for the following reasons:

- How to optimize is not clear. What methods have been used for sample generating and what algorithms have been used. A complete section with details is necessary for this subject;

We thank the reviewer for this comment. We now provide in the results section, a new paragraph on the rationale, optimization and methods used to generate the new compounds

- Why the authors did not add keywords?!;

We thank the reviewer 4 for this remark and the keywords are now added in the revised manuscript under the abstract

- Figures 3,4,6,7,8,10,11 and 12 are not clear and they are hard to read;

We thank the reviewer for this remark; all the figures are now provided in good format (TIF) with higher resolution (600 dpi).

- Generally, there are mistakes in some sentences of the manuscript. So related corrections should be applied; 

We thank the reviewer for this comment. We have asked the editor to use the English editing service of the journal to this issue.

- The References section has not been well organized.

The references are now updated as per journal requirements.

Round 2

Reviewer 3 Report

Authors revised manuscript as per suggestions. Current version can be considered for publication.

Reviewer 4 Report

Authors have addressed my questions..